# Actual Scope of Nursing Practice in Saudi Ministry of Health Hospitals

**DOI:** 10.3390/healthcare12070785

**Published:** 2024-04-04

**Authors:** Ahmed A. Hamadi, Ahmad E. Aboshaiqah, Naif H. Alanazi

**Affiliations:** 1Nursing Services Department, Oyun City Hospital, Alahsa Health Cluster, Alahsa 14889, Saudi Arabia; aahamadi@moh.gov.sa; 2Nursing Administration and Education Department, College of Nursing, King Saud University, Riyadh 12372, Saudi Arabia; aaboshaiqah@ksu.edu.sa; 3Medical-Surgical Nursing Department, College of Nursing, King Saud University, Riyadh 12372, Saudi Arabia

**Keywords:** actual scope of practice, cross-sectional design, nursing, Saudi Arabia

## Abstract

Background: Articulating a clear scope of practice for all nursing categories is essential for improving patient safety, quality of care, and nurse retention. However, this is not the case in many countries, including Saudi Arabia. This study aimed to analyze the actual scope of practice for nursing staff in Saudi Ministry of Health hospitals. Methods: The study used a cross-sectional exploratory design. The sampling method used in the study was the quota sampling technique. The scale utilized in this study was the Actual Scope of Practice (ASCOP) scale. Data were collected in March 2021 through an electronic form questionnaire completed by 286 nurses in two hospitals in Al-Hasa province in Saudi Arabia. Results: The overall mean score for ASCOP was 4.64 out of 6. When participants were grouped according to select characteristics (various nursing staff categories, educational levels, years of experience, nationality, gender, and type of work setting), the results revealed no statistically significant differences in overall ASCOP mean scores, except for gender and nationality. Conclusions: The overall mean scores of nursing activities performed in practice do not significantly differ across nurses with different professional categories (health assistant, nursing technician, nursing specialist, and senior nursing specialist), indicating no clear scope of practice for each nursing category, in turn leading to role overlap among them in practice. The current study’s findings can guide decision-makers to develop a clear scope of practice for nurses. The findings should also urge the decision-makers to reevaluate the usefulness of having multiple professional categories of nurses who are allowed to carry out almost the same job duties.

## 1. Introduction

Given that nursing professionals represent the largest percentage of healthcare workers in health institutions, they play a crucial role in enhancing patient safety and the quality of care [1]. However, several studies have shown that nursing staff members were utilized improperly; either they were overutilized (carrying out skills beyond their qualification and experience) or underutilized (not practicing to their full scope of practice), in turn negatively impacting the cost-effectiveness, quality of care, patient safety, and job satisfaction [2,3,4].

The scope of nursing practice can be broadly defined by the Queensland Nursing Council (QNC, p. 9) as “that which nurses are educated, authorized and competent to perform” [5]. Similarly, the Nursing and Midwifery Board of Ireland (NMBI) defines the scope of practice for a registered nurse as “the range of roles, functions, responsibilities and activities which a registered nurse is educated, competent and has authority to perform” [6] (p. 7). However, several studies [3,7,8,9] reported a gap between the optimal or full scope of practice for nurses, which is defined as “the competencies developed in educational programs and the acts permitted to them by legislation” [3] (p. 266). It was also defined as “a role that is reflected in the knowledge base of the profession” [7] (p. 6), on the one hand, and their actual (enacted) scope of practice, which is defined as “the range of functions and responsibilities carried out by nurses in their daily work” [3] (p. 57). D’Amour et al. [10] developed the first tool for measuring the enacted (actual) scope of practice for nurses working in the hospital called the Actual Scope of Practice (ASCOP) questionnaire, by which nursing leaders and researchers can assess to what extent the activities that performed by nursing are consistent with their professional preparation.

Several studies highlight that nurses are not working to a full scope of practice that is consistent with their education and training [7,10,11]. Déry et al. [3] studied 335 nurses using the ASCOP questionnaire. The average score of participants was 3.21 out of 6, indicating that nurses occasionally carry out activities related to the enactment of the nursing scope of practice [3]. Furthermore, a study conducted in the United States by Nathenson et al. [12] reported that nursing staff spend only 38% of their time carrying out essential nursing roles.

A mixed-method study from Canada also revealed that the majority (75%) of registered psychiatric nurses (RPNs) reported not being appropriately utilized, more than 80% of licensed practical nurses (LPNs) reported that they were not working to full scope, and about 50% of registered nurses (RNs) reported that they were not fully utilized [9]. This issue may be attributed to internal factors such as feelings of incompetence and personal characteristics (e.g., education level) [13,14] or external factors such as organizational support, work environment, institutional policies, and national health policy [8,11,13,14,15]. Not allowing nursing staff to work their full scope of practice negatively impacts the quality of care, and it is associated with job dissatisfaction and disappointment among nurses who feel that they have been overeducated [10,13,16].

Braithwaite [17] studied 178 nurses in primary healthcare institutions in Canada, using the ASCOP questionnaire with some adjustments to be appropriate for measuring the scope of nursing practice in primary healthcare institutions. The results revealed that the average score of subjects was 5.16 out of 6, indicating that nurses almost always carry out activities related to the enactment of nursing scope of practice [17]. An exploratory study conducted in rural areas with low medical services in Tanzania also reported that nurses performed activities beyond their scope of practice, such as prescribing medication and carrying out minor surgical procedures [18]. A study by Younan et al. [2] in Lebanon showed that nursing aides were engaging in activities beyond their scope of practice.

A number of studies showed that the characteristics of nursing staff and the area where they work significantly influence their actual scope of practice [3,13,19,20]. For instance, the work of Irvine et al. [20] indicated that a nurse’s education level, experience, skills, and workplace characteristics influence the ability to engage in nursing roles effectively in independent roles (e.g., nursing assessment, diagnosis, and intervention) and dependent roles (e.g., carrying out physician’s orders), as well as interdependent roles (e.g., communication, coordination of care, and case management). A study conducted in Quebec, Canada, revealed a strong relationship between education level and enacted (actual) scope of practice; that is, nurses with baccalaureate education levels (or higher) performed more complex tasks than nurses with lower education levels [13].

Research by Déry et al. [3], in a pediatric university hospital in Canada, aimed to examine the relationship between the enacted (actual) scope of nursing practice on the one hand and education level and position on the other, revealing that nurses who hold bachelor’s degrees had a significantly greater actual scope of practice than nurses who hold diploma degrees. This assumption is supported by ICN [19], which reported that a nurse’s education level is one of the significant factors that affect the scope of practice, in addition to other factors such as the years of experience and practice area. However, a cross-sectional survey of 2852 nurses working in 39 hospitals in Lebanon that aimed to investigate the actual scope of practice for various nursing categories showed that it was associated positively with the years of experience, whereas no differences were found among nurses with various educational levels [2].

Articulating a clear scope of practice for all nursing categories is essential for improving patient safety, quality of care, and nursing retention [2]. However, this is not the case in many countries. For example, in Australia, an integrative review conducted by Birks et al. [21] reported that Australia’s scope of nursing practice is not stated obviously. Consequently, the roles of registered nurses and enrolled nurses overlap. In Canada, White et al. [9] showed that blurring exists around various nursing categories’ roles. Furthermore, in Lebanon, Younan et al. [2] reported that the boundaries between the scope of practice of registered nurses and practical nurses are ambiguous. Shuriquie et al. [22] studied 348 nurses (staff nurses and practical nurses) from three healthcare sectors in Jordan (government, army, and private), using a cross-sectional questionnaire. The result revealed that the role boundary between a staff nurse and a practical nurse is clear [22].

In Saudi Arabia, the Nursing Board was initiated in 2002 under the direction of the Saudi Commission for Health Specialties (SCFHS) [23], aiming mainly to define the members of the profession and the profession itself; identify the nursing scope of practice; establish standards for nursing education, ethics, and practice; and develop accreditation and accountability systems [24]. Although the Nursing Board succeeded in developing standards of registration and licensure in 2003, setting standards for training and education activities, and creating a code of ethics, it failed in articulating the scope of practice for various nursing categories (technicians, specialists, consultants) [25]. A previous study conducted in 2019 that used the Arabic version of ASCOP established that there was a range of variations in the scope of practice among nurses in Saudi Arabia [24]. Consequently, in practice, their roles may overlap and be confused. In light of other findings, the present study investigated the nursing staff’s actual scope of practice in the Ministry of Health (MOH) hospitals in Saudi Arabia and provided the Nursing Board with a baseline description of it.

As the healthcare system has become more complex and nursing roles developed significantly, the study of the enacted scope of nursing practice has become necessary to meet these changes [2]. Therefore, numerous studies have been conducted worldwide. In Saudi Arabia, Al-Dossary [25] clarified that the nursing scope of practice is not articulated yet. Studying the actual nursing scope of practice is, therefore, necessary to guide and regulate the profession. According to the International Council of Nurses (ICN), the scope of practice is significant because it forms the basis for compiling standards of practice, education curriculums, and job descriptions, and for protecting the nursing profession through a legal framework that indicates the required qualifications and authorities for providing nursing services and interventions [19]. The scope of practice also distinguishes nurses from other health professionals and may prevent scope creep from other health cadres [26]. It can also considerably impact the quality of care and welfare of nurses. In this study, we sought to provide the Nursing Board—a professional organization, one of the duties of which is identifying the nursing scope of practice—with a baseline description of the actual scope of practice for nurses in Saudi Arabia. This study aimed to analyze the actual scope of practice of nursing staff in hospitals of the Saudi MOH. This aim was achieved through the following objectives, to (1) determine the actual scope of nursing practice among staff members working in the Saudi MOH hospitals, and (2) identify whether differences or similarities exist in the actual scope of practice among nursing staff concerning their professional categories and educational levels.

## 2. Materials and Methods

### 2.1. Design and Settings

This study used a cross-sectional exploratory design. This design is the appropriate method for the current study to describe the phenomenon and the actual scope of nursing practice and explore the associations and differences between the collected variables [27]. The present study surveyed nursing staff in two government hospitals operated by the MOH in Al-Hasa province, Saudi Arabia: King Fahad Hospital Hofuf (KFHH) and Prince Saud Bin Jalawi Hospital (PSBJH). Participants were recruited from the following units: medical, surgical, pediatric, adult intensive care, pediatric intensive care, isolation, and burns.

### 2.2. Sample and Sampling Method

The target population was all nursing personnel working at MOH hospitals in Saudi Arabia. The accessible population for this study included the nursing staff working in the two hospitals of KFHH and PSBJH. The total number of nursing staff in KFHH was 721 and in PSBJH was 285. Thus, the accessible population was equal to 721 + 285 = 1006. The minimum sample size required was calculated utilizing Slovin’s formula, n = N/(1 + N*e^2^), where n = sample size, N = population, and e = margin of error (5%), resulting in a sample size of 286 participants.

The sampling method used in the study was the quota sampling technique. First, the researchers chose the relevant stratification and divided the population accordingly: KFHH (721) and PSBJH (285). Then, they calculated the quota for each stratum as follows: KFHH represented 72% of the accessible population. Therefore, 72% of the sample size (286) was the quota of KFHH = 0.72 × 286 = 205.92, or ~206. PSBJH represented 28% of the accessible population. Thus, 28% of the sample size (286) was the quota of PSBJH = 0.28 × 286 = 80.08, or ~80. Then, the researchers continued to invite participants until each stratum’s quota was met. Participants were required to meet the following inclusion criteria: (a) be a registered nurse working in a technical position, working in-patient units, and in direct patient care; (b) have completed at least six months of service (to ensure those study participants have at least some familiarity with the job and organization); and (c) be willing to participate.

### 2.3. Data Collection and Materials

Data were collected in March 2021 utilizing a self-administered questionnaire. To conduct a pilot study in both hospitals (KFHH and PSBJH), we asked the in-patient units’ head nurses to send the electronic questionnaire to a small number of staff who met the inclusion criteria. After completing the pilot study, the head nurses were asked to send the electronic (online) questionnaire form to all individuals who met the inclusion criteria. The participants were asked to select the ‘Yes’ option if they willingly and voluntarily agreed to participate in the survey and, otherwise, to select the ‘No’ option. We continued receiving responses until the quota for both hospitals was met.

The ASCOP questionnaire, developed by D’Amour et al. [10], was utilized in this study to collect information on the actual scope of nursing practice in the two hospitals (KFHH and PSBJH) operated by MOH, Saudi Arabia. ASCOP comprises 26 items grouped into six dimensions: (1) assessment and care planning, with five items, (2) the teaching of patients and families, with four items, (3) communication and care coordination, with five items, (4) integration and supervision of staff, with four items, (5) quality of care and patient safety, with five items, and (6) knowledge updating and utilization, with three items. The items are measured on a six-point Likert-type scale (1: never; 2: very rarely; 3: sometimes; 4: frequently; 5: almost always; 6: always). The items in the ASCOP questionnaire were developed corresponding to the three levels (levels 1 to 3) of complexity of activities. Level 1 corresponded to a low level of complexity, level 2 as a moderate level of complexity, and level 3 as a high level of complexity. Each item was translated into the Arabic language using the Arabic translation of the ASCOP by Younan et al. [28] to be suitable for those who were weak in English.

In our study, the instrument showed acceptable internal consistency with an overall Cronbach’s alpha of 0.93 and Cronbach’s alpha ranging from 0.66 to 0.86 for the six dimensions. The psychometric assessment carried out by D’Amour et al. [10] also found this instrument valid and reliable, with an overall Cronbach’s alpha of 0.89 and Cronbach’s alpha for the six dimensions ranging from 0.61 to 0.70. The instrument has been further translated by Younan et al. [28], with an overall Cronbach’s alpha of 0.96.

### 2.4. Ethical Considerations

Our study was approved by the Institutional Review Board of Alahsa Health Cluster, Ministry of Health, Saudi Arabia (protocol code: H-05-HS-065, dated 22 November 2020). Ethical Committee approval was also obtained from the study hospitals. Permission to use the ASCOP tool was obtained from the ASCOP author. Furthermore, permission to use the Arabic translation of the ASCOP tool was obtained from the translator. In the first part of the online survey, the purpose of the study and their voluntary participation was explained to the participants. The participants were also assured about their right to withdraw from the study at any time without any consequence related to their work. The survey was anonymous, with no identifying information collected. The data collected in this study were saved in a secured file and kept confidential with encryption and password, to which only the research team had access.

### 2.5. Data Analysis

All statistical analysis procedures were performed using SPSS 20. Descriptive statistics involving frequency, percentages, means, and standard deviations (SDs) were calculated and presented in tables to describe and summarize the study data. Independent sample *t*-tests and one-way analysis of variance (ANOVA) with post hoc tests (Bonferroni/Games–Howell) were used to find the statistical differences between ASCOP mean total and subscale scores across participants’ characteristics. In all analyses, *p* < 0.05 was considered significant.

## 3. Results

### 3.1. Demographic Characteristics of Participants

The total number of questionnaires distributed was 380. The number of questionnaires returned was 291, resulting in a response rate of 76.6%. Two incomplete questionnaires and three invalid responses were excluded, leaving 286 participants in this study. The majority of the participants were female (89.2%), non-Saudi (57%), held a bachelor’s degree (68.9%), and were registered nurses who were working as nursing specialists (54.9%). The highest proportion of the participants were working in adult intensive care units and had 5–10 years of working experience. Participants’ demographic characteristics data are presented in Table 1.

### 3.2. Actual Scope of Nursing Practice in Saudi Ministry of Health Hospitals

As Table 2 shows, the overall mean score for ASCOP is 4.64 out of 6. The mean scores of dimensions ranged from 4.29 to 4.80. The most frequently carried out activities were those concerning assessment and care planning at 4.80, and the teaching of patients and families at 4.80, followed by those related to the quality of care and patient safety at 4.75, knowledge updating and utilization at 4.64, and communication and care coordination at 4.54. The least-performed activities were those related to integration and supervision of staff at 4.29.

### 3.3. Actual Scope of Practice across Nurses’ Professional Categories

As shown in Table 3, the overall mean score on ASCOP was highest among senior nursing specialists, followed by nursing specialists, nursing technicians, and health assistants. However, those findings were not statistically significant (*p* = 0.209). Looking at the dimension scores, four out of the six dimensions of nursing activities did not significantly differ across nursing specialists, nursing technicians, and health assistants—teaching patients and family, communication and care coordination, integration and supervision of staff, and quality of care and patient safety: *p* = 0.472, 0.486, 0.857, and 0.184, respectively. The results of the other two dimensions, assessment and care planning (*p* = 0.024) and knowledge updating and utilization (*p* = 0.018), revealed significant differences. Post hoc tests (data not presented in Table 3) were further calculated between those categories, which indicated that nursing specialists have significantly higher mean scores than nursing technicians in assessment and care planning (*p* = 0.015). Senior nursing specialists have significantly higher mean scores than nursing technicians in knowledge updating and utilization (*p* = −0.027).

### 3.4. Actual Scope of Practice across Nurses’ Educational Levels

The nurses with a postgraduate degree have the highest overall mean score on ASCOP, followed by nurses with a bachelor’s degree and nurses with a diploma degree. However, those results were not statistically significant (*p* = 0.202), as shown in Table 4. Regarding dimension scores, all dimensions of nursing activities did not significantly differ across nurses with different educational levels (postgraduate, bachelor, and diploma degrees) except for the assessment and care planning dimension. A significant difference was only found between nurses with a bachelor’s degree and those with a diploma degree (*p* = 0.019), where the post hoc test results (data not presented in Table 4) revealed that bachelor’s degree nurses have significantly higher mean scores than diploma nurses in nursing activities pertaining to assessment and care planning (*p* = 0.004).

### 3.5. Actual Scope of Practice across Various Participants’ Characteristics

As regards the years of experience, some significant differences were found in the two subscales. Specifically, nurses with more than 25 years of experience showed significantly higher mean scores on the teaching of patients and family activities than those with less experience (<5 years, 5–10 years, or 11–25 years of experience at *p* = 0.041, 0.040, and 0.003, respectively). Furthermore, nurses with 5–10 years of experience scored significantly higher on integration and supervision of staff activities than those with less than 5 years of experience (*p* = 0.003).

Except for integration and supervision of staff, female nurses have a significantly higher mean score than male nurses on the overall ASCOP (*p* = 0.006) and all the individual subscales (assessment and care planning at *p* = 0.001, the teaching of patients and family at *p* = 0.007, communication and care coordination at *p* = 0.019, quality of care and patient safety at *p* = 0.009, and knowledge updating and utilization at *p* = 0.012). Significant dimension score differences by nationality were observed in assessment and care planning (*p* = 0.005), communication and care coordination (*p* = 0.015), quality of care and patient safety (*p* = 0.001), knowledge updating and utilization (*p* = 0.001), and overall ASCOP (*p* = 0.005), in which non-Saudi nurses were found to perform the activities related to these dimensions more frequently than Saudi nurses. No statistically significant variances in the overall ASCOP and subscales were identified across nurses from different nursing units (Table 5).

## 4. Discussion

The overall ASCOP mean score (4.64 out of 6) in this study is higher than in a number of previous studies utilizing the same instrument. For example, a Canadian study conducted by Déry et al. [3] had an overall mean score for ASCOP of 3.21 out of 6, indicating that, on average, nurses occasionally performed nursing skills measured by the ASCOP scale. In another study carried out in Canada, a higher overall item mean score of 4.8 (out of 6.0) was reported [17]. Furthermore, D’Amour et al. [14] conducted a study where the overall mean score for the ASCOP was 3.47 out of 6. A study by Younan et al. [2] in Lebanon showed that the overall ASCOP mean score was 4.42/6. Compared with the results of these studies, the present study results showed that nurses working in the Saudi MOH hospitals assumed broader roles and had a higher ASCOP mean score (4.64 out of 6). Our overall mean score is also higher than a previous Saudi study [24], but the comparison must be considered with caution as the study was conducted in 2005 [24]. This result may be attributed to the current direction of the Saudi government to develop and empower the nursing profession as one aim of Saudi Vision 2030 by establishing a clear scope of nursing practice guidelines in the country [25,28].

Among the six dimensions of nursing activities, the most frequently carried out were those about assessment and care planning and the teaching of patients and families. This finding was consistent with those of two studies conducted in Canada, which found that these two dimensions had the highest mean scores among the six items in the ASCOP scale [3,10]. In other studies, nurses in other countries including Hong Kong, Canada, Sweden [29], and in the United States of America [30] have reported that involving patients and their families in nursing care is important to advance the practice of patient- and family-centered nursing care. Nursing assessment is the first step in the nursing process, and the accuracy of the other four steps depends heavily on the completeness and correctness of the data collected in this step [31,32,33,34]. This may explain the results of the current study, which found that the activities related to assessment and care planning were the most frequent nursing activities performed. A study conducted by Younan et al. [2] showed that teaching patients and families was the most frequently performed activity as reported by participants, which is consistent with the results of the current study. However, assessment and care planning ranked fifth, which is inconsistent with the current study.

According to Dumit [35], the teaching of patients and families is a central role of nursing, and the majority of patient and family education in any health institution is provided by nurses. In this respect, nurses should train the patient and family about the main aspects, mechanisms, benefits, and potential complications of each treatment procedure [35]. Nurses should also educate the patient and family before discharge from the health institution focusing on all information needed about medications, lifestyle modifications, self-monitoring for signs and symptoms of disease, self-monitoring devices (e.g., glucose meter and blood pressure monitor), and self-care, such as teaching patients how to inject insulin and how to bathe an infant [35]. This role necessitates nurses allocating enough time to carefully determine the educational needs, planning appropriate education, providing it at the right time and place, and evaluating its outcomes. This may be used to interpret the high score of the teaching patients and families’ dimension in the current study and many previous studies [2,3,10].

The nursing activities related to integration and supervision of staff (such as participating in identifying the in-service education needs of the unit, orientation, and training of newly hired nursing staff or nursing students, developing and conducting training activities for the care team, and acting as an educator or mentor for newly hired nursing staff) have the lowest mean score, in line with the results of the previous studies [2,3,10]. This finding indicates that, on average, the nurses in the sample have low involvement in planning and developing unit education and training programs and acting as mentors or educators for other staff, such as newly hired nursing staff and nursing students.

Our findings also showed no significant difference in overall ASCOP scores across nurses with different educational levels, as found in many previous studies [2,10,17]. In other words, in practice, nurses at three different educational levels (diploma degree, bachelor’s degree, and postgraduate degree) perform similar nursing activities with the same level of complexity, potentially negatively impacting the nursing staff’s willingness to seek higher education. This assumption is supported by two studies [2,36] that revealed that the number of nurses with a postgraduate degree increases in the regions that empower them to practice to their full scope (performing nursing activities that are different and more complex than that performed by nurses with a lower degree, such as a bachelor’s degree).

However, this finding contrasts with several studies [3,13,19,20,37], which found that nurses’ education level significantly influences their actual scope of practice. Déry et al. [3] revealed that nurses who hold bachelor’s degrees had a significantly greater actual scope of practice than nurses who hold diploma degrees. A subsequent study conducted by Déry et al. [13] also found that nurses with baccalaureate education levels (or higher) performed more complex tasks than nurses with lower education levels.

Contrary to many studies conducted in different countries [10,17,19,20,22], our results revealed that the actual scope of nursing practice was similar across nurses in different professional categories. This finding argues the decision-makers, especially in the MOH, about the usefulness of having multiple professional categories of nurses (e.g., health assistant, technician, specialist, senior specialist) that are allowed to carry out almost the same job duties. Another important finding in this study is that the actual scope of practice for non-Saudi nurses was significantly broader than that of Saudi nurses, which is aligned with the results of a study conducted by Alabdulaziz et al. [38]. One of the possible reasons might be the poor English language skills of many Saudi nurses. This assumption is supported by two studies [39,40] that revealed that Saudi nurses have some challenges in providing good care. One of these challenges is mastering English proficiency.

### Limitations

The current study has some limitations, as follows: (1) the current study’s data were collected through a self-administered questionnaire, which is susceptible to the risk of different response biases such as social desirability response bias; (2) the sample of nurses involved in the survey was not drawn randomly, which may bias the findings; and (3) the study was conducted only in two MOH hospitals in one geographic area of Saudi Arabia.

## 5. Implications

The findings of this study can guide decision-makers (the leaders in the nursing board in SCFHS and the leaders in the Saudi MOH) in developing a clear scope of practice for nurses, which, in turn, can form the basis for compiling standards of practice, curriculums, and job descriptions. This is also a basis for protecting nurses through a legal framework that specifies who is qualified and authorized to provide certain services and interventions and distinguishes nurses from other health professionals. This may consequently prevent scope creep from other health cadres. Ultimately, it will considerably impact the quality of care and welfare of nurses.

The findings of the current study should urge decision-makers, especially in the MOH, to reevaluate the usefulness of having multiple professional categories of nurses (e.g., health assistants, technicians, specialists, and senior specialists) who are allowed to carry out almost the same job duties. The results of the present study should urge the nursing leaders to address the low involvement of nursing staff in the activities related to integration and supervision of staff, such as identifying in-service education needs for their department, orienting and training newly hired nurses or nursing students, acting as an educator or mentor for newly hired nurses, and developing and conducting training activities for the care team, in accordance with their skills.

## 6. Conclusions

Articulating a clear scope of practice for all nursing categories is essential for improving patient safety, quality of care, and nursing retention. However, the findings of this study investigating the actual scope of nursing practice in the MOH’s hospitals in Saudi Arabia revealed no statistically significant differences in the overall mean scores of nursing activities performed in practice across nurses in different professional categories (health assistant, nursing technician, nursing specialist, and senior nursing specialist). This indicates that no clear scope of practice exists for each nursing category, in turn leading to role overlap among them in practice.

## Figures and Tables

**Table 1 healthcare-12-00785-t001:** Demographic Characteristics of Participants.

Sample Characteristics	n	%
Hospital:		
King Fahad Hospital-Hofuf	206	72
Prince Saud Bin Jalawi Hospital	80	28
Gender:		
Male	31	10.8
Female	255	89.2
Nationality:		
Saudi	123	43
Non-Saudi	163	57
Level of education:		
Diploma degree	83	29
Bachelor’s degree	197	68.9
Postgraduate degree	6	2.1
Years of experience:		
<5 Y	112	39.2
5–10 Y	118	41.3
11–25 Y	54	18.9
>25 Y	2	0.7
Professional category:		
Health Assistant	17	5.9
Nursing Technician	107	37.4
Nursing Specialist	157	54.9
Senior Nursing Specialist	5	1.7
Type of unit:	5	1.7
Medical	92	32.2
Surgical	45	15.7
Pediatric	18	6.3
Adult intensive care	94	32.9
Pediatric intensive care	11	3.8
Isolation	12	4.2
Burns	14	4.9

Note. *N* = 286.

**Table 2 healthcare-12-00785-t002:** Mean Scores of ASCOP Scale’s Dimensions.

Dimension	Mean	SD
Assessment and care planning	4.80	0.89
Teaching of patients and families	4.80	0.99
Communication and care coordination	4.54	1.06
Integration and supervision of staff	4.29	1.22
Quality of care and patient safety	4.75	0.99
Knowledge updating and utilization	4.64	0.96
Overall	4.64	0.90

Note. *N* = 286. ASCOP = Actual Scope of Practice Scale.

**Table 3 healthcare-12-00785-t003:** Scores on ASCOP Scale’s Dimensions by Professional Categories.

Dimension	Health Assistant		Nursing Technician		Nursing Specialist		Senior Nursing Specialist		*p* ^a^
	M	SD	M	SD	M	SD	M	SD	
Assessment and care planning	4.29	1.35	4.63	0.94	4.96	0.76	4.92	0.27	0.024 *
Teaching of patients and family	4.60	1.27	4.70	0.99	4.89	0.96	4.80	0.48	0.472
Communication and care coordination	4.55	1.52	4.41	1.06	4.61	1.01	4.72	0.69	0.486
Integration and supervision of staff	4.25	1.29	4.27	1.20	4.29	1.24	4.75	0.50	0.857
Quality of care and patient safety	4.75	1.34	4.62	0.97	4.83	0.96	5.08	0.44	0.184
Knowledge updating and utilization	4.55	1.18	4.55	0.94	4.70	0.96	5.13	0.30	0.018 *
Overall mean on ASCOP	4.50	1.24	4.53	0.9	4.73	0.87	4.89	0.36	0.209

Note: *N* = 286. ASCOP = Actual Scope of Practice Scale, *** significance at 0.05. ^a^ Significance levels were calculated using a one-way analysis of variance across the four categories.

**Table 4 healthcare-12-00785-t004:** Scores on ASCOP Scale’s Dimensions by Educational Levels.

Dimension	Diploma		Bachelor		Postgraduate		*p* ^a^
	M	SD	M	SD	M	SD	
Assessment and care planning	4.51	1.01	4.92	0.81	4.73	0.52	0.019 *
Teaching of patients and family	4.67	1.05	4.86	0.97	4.67	0.54	0.291
Communication and care coordination	4.38	1.12	4.60	1.04	4.57	0.72	0.296
Integration and supervision of staff	4.28	1.19	4.28	1.24	4.71	0.46	0.141
Quality of care and patient safety	4.61	1.06	4.81	0.96	4.87	0.65	0.301
Knowledge updating and utilization	4.51	0.95	4.69	0.97	4.94	0.53	0.179
Overall mean on ASCOP	4.49	0.95	4.70	0.89	4.74	0.50	0.202

Note: *N* = 286. ASCOP = Actual Scope of Practice Scale, *** significance at 0.05. ^a^ Significance levels were calculated using a one-way analysis of variance across the three categories.

**Table 5 healthcare-12-00785-t005:** Scores on ASCOP Scale’s Dimensions across Various Participants’ Characteristics.

Sample Characteristics	Overall Mean on ASCOP	Assessment and Care Planning	Teaching of Patients and Family	Communication and Care Coordination	Integration and Supervision of Staff	Quality of Care and Patient Safety	Knowledge Updating and Utilization
Type of unit							
Medical	4.68	4.82	4.83	4.50	4.50	4.73	4.66
Surgical	4.64	4.73	4.83	4.60	4.24	4.75	4.61
Pediatric	4.87	4.81	5.07	4.72	4.65	4.99	5.04
ICU	4.55	4.75	4.65	4.51	4.06	4.69	4.55
PICU	4.85	5.16	5.14	4.64	4.14	5.09	4.88
Isolation	4.16	4.52	4.40	3.92	3.71	4.27	4.11
Burns	5.05	5.14	5.32	4.99	4.68	5.13	4.98
*p*-values	0.174	0.448	0.092	0.266	0.056	0.267	0.110
Gender							
Male	4.23	4.20	4.35	4.12	4.10	4.32	4.30
Female	4.69	4.87	4.86	4.59	4.31	4.81	4.68
*p*-values	0.006 **	0.001 ***	0.007 **	0.019 *	0.378	0.009 **	0.012 *
Years of experience							
<5 Y	4.52	4.77	4.76	4.42	3.98	4.62	4.50
5–10 Y	4.78	4.92	4.89	4.65	4.53	4.88	4.77
11–25 Y	4.60	4.58	4.68	4.57	4.40	4.71	4.62
>25 Y	5.00	5.40	5.88	3.80	4.13	5.60	5.33
*p*-values	0.158	0.261	0.001 ***	0.291	0.006 **	0.136	0.134
Nationality							
Saudi	4.47	4.63	4.70	4.36	4.14	4.54	4.42
Non-Saudi	4.77	4.93	4.88	4.67	4.40	4.91	4.81
*p*-values	0.005 **	0.005 **	0.140	0.015 *	0.077	0.001 ***	0.001 ***

Note: *N* = 286. ICU = adult intensive care unit; PICU = pediatric intensive care unit. *** Significance at 0.05; ** significance at 0.01; *** significance at 0.001.

## Data Availability

The data presented in this study are available on request from the corresponding author.

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
