# Peer review of "Actual Scope of Nursing Practice in Saudi Ministry of Health Hospitals"

_healthcare, 2024, doi:10.3390/healthcare12070785_

Round 1

Reviewer 1 Report

Comments and Suggestions for Authors

The topic of the manuscript is interesting. The Aim of the Study is well set, it is divided into two essential parts. This manuscript has a contribution of great importance, in many countries of the world there is no clear description of nurses with different levels of education. In this manuscript, the set goal has been achieved. Data analysis is well done. The tables are clear to interpret. In line 317, what are Saudi Arabia's goals for 2023. If the reader does not know, it would be good to put them in parentheses up to two sentences. Row 320 only compare to Canada, is there any other country to compare your results with? The discussion is extremely well written. There are a lot of problems with this topic in Europe. Nurses who graduate from university and who have postgraduate studies do the same job with no difference in salary or professional status. I believe that this work has started the eternal theme of harmonizing the status, job description and provision of quality health services in relation to education. The conclusion is clear and based on evidence. Most of the references are of an older date, I think it would be desirable to refresh them with references not older than five years.

Author Response

REVIEWER 1 Comments

The topic of the manuscript is interesting. The Aim of the Study is well set, it is divided into two essential parts. This manuscript has a contribution of great importance, in many countries of the world there is no clear description of nurses with different levels of education. In this manuscript, the set goal has been achieved. Data analysis is well done. The tables are clear to interpret.

RESPONSE: Thank you very much for the positive comments. We hope these will help convince the editor to consider accepting our paper for publication.

In line 317, what are Saudi Arabia's goals for 2023. If the reader does not know, it would be good to put them in parentheses up to two sentences.

RESPONSE: We added this, as suggested. Thank you.

Row 320 only compare to Canada, is there any other country to compare your results with?

RESPONSE: We added two comparisons from different countries. Thank you.

The discussion is extremely well written. There are a lot of problems with this topic in Europe. Nurses who graduate from university and who have postgraduate studies do the same job with no difference in salary or professional status. I believe that this work has started the eternal theme of harmonizing the status, job description and provision of quality health services in relation to education. The conclusion is clear and based on evidence.

RESPONSE: Thank you very much for the positive comments. We hope these will help convince the editor to consider accepting our paper for publication.

Most of the references are of an older date, I think it would be desirable to refresh them with references not older than five years.

RESPONSE: We updated the references, not older than five years, as recommended. Thank you.

Reviewer 2 Report

Comments and Suggestions for Authors

Comments on the Quality of English Language

 English language is good. Minor editing of English language required

Author Response

Reviewer 2 Comments

General comments. The paper is well structured and developed. However, some integrations/clarifications/definitions seem needed, especially in Materials/Methods and Results sections.

RESPONSE: Thank you very much for the positive comment, and other comments for the improvement of our paper.

Specific comments

Abstract. The following sentence is not fully clear “The results revealed no statistically significant differences in overall ASCOP mean scores when participants were grouped according to select characteristics (various nursing staff categories, educational levels, years of experience, nationality, gender, and type of working setting) except for gender and nationality”. It could be rewritten as follows: “When participants were grouped according to select characteristics (various nursing staff categories, educational levels, years of experience, nationality, gender, and type of working setting), the results revealed no statistically significant differences in overall ASCOP mean scores, except for gender and nationality”.

RESPONSE: We have rewritten this sentence, as suggested. Thank you very much.

Keywords. I would suggest to delete “scope of practice” (Authors already indicated “actual scope of practice”), and add “cross-sectional design”

RESPONSE: We have deleted “scope of practice” and added “cross-sectional design”.

Introduction. This section is well done. I would however suggest the following:

- overall, lines 33-53, and par. 1.1. e 1.2. could be merged. In particular, lines 33-53 anticipate some aspects that are repeated also below. Moreover, par. 1.2. could be integrated (without sub-paragraphs) at the end of the Introduction (current par. 1.1.);

RESPONSE: We have revised this, as suggested. Thank you very much.

- line 67: the number page of the citation is lacking;

RESPONSE: The definition of optimal scope of practice for nurses was defined only in page 266 of the reference: Déry, J., Clarke, S. P., D’Amour, D., & Blais, R. (2016). Education and role title as predictors of enacted (actual) scope of practice in generalist nurses in a pediatric academic health sciences center. JONA: The Journal of Nursing Administration, 46(5), 265–270. So, we did not do any revision about this comment. Please clarify and thank you.

- lines 77-80: the description of ASCOP is also presented in the Methods section, thus it could be deleted from the Introduction, or at least leave a brief summary there;

RESPONSE: We decided to delete ASCOP description in the Introduction.

- lines 88-90: the following full definitions are lacking: Licensed Practical Nurses (LPNs), Registered Nurses (RNs) and Registered Psychiatric Nurses (RPNs).

RESPONSE: Since this is the only part the abbreviations appear in the paper, we decided to write their full definitions.

Materials and Methods. This section includes all relevant aspects, but I would suggest some adjustments and integrations:

- to combine par. 2.1 Design and par. 2.2 Setting, and also provide more information on the cross-sectional exploratory design. Moreover, a reference could be added in this respect;

RESPONSE: We revised these parts, as suggested, and updated the numbering of sub-headings.

- to combine par. 2.4 Data Collection and par. 2.5 Materials, and also explain how participants were informed;

RESPONSE: We revised these parts, as suggested. Thank you.

- par. 2.5: more infos on ASCOP are needed, e.g., how many items for each dimension, score range, what high and low values indicate e.g. low = nurses occasionally performed nursing skills measured by the ASCOP scale (as Authors indicate at lines 309-310). Also, the scale could be included as Supplementary material;

RESPONSE: We revised this part, as suggested. We added the levels of complexity for ASCOP. Thank you.

- line 195: D'Amour et al. is ref. 14 and not 12;

RESPONSE: This has been corrected. Thank you.

- lines 205-206: Authors could specify “in our study”;

RESPONSE: We added this phrase. Thank you.

- par. 2.6: it is useful to include also here what is stated at lines 419-420 (study approved by the Institutional Review Board of Alahsa Health Cluster, Ministry 419 of Health, Saudi Arabia, protocol code: H-05-HS-065 and November 22, 2020). Authors could also describe more the informed consent (signed online?), and how privacy/anonymity/confidentiality of personal information collected were assured;

RESPONSE: We revised this section of the Methods, as suggested. Thank you.

- par. 2.7: Authors mention pie charts and bar graphs for presenting the results, but I cannot find them in the manuscript;

RESPONSE: The ‘pie charts and bar graphs’ have been deleted as they were typographical errors. Thank you.

- line 219: Solvin’s formula is already mentioned in par. 2.3.;

RESPONSE: We also deleted this sentence. Thank you.

- lines 219-220: the sentence on Cronbach’s alpha is already included in par. 2.5.

RESPONSE: We also deleted this sentence. Thank you.

Results. On the whole, this section is exhaustive, but data are almost presented as lists. I would suggest, for instance the following:

- to reduce par. 3.1. and indicate only the main characteristics of the sample (other findings can be found in the tables), and to put in the text only % and not absolute values (to simplify the reading);

RESPONSE: This part has been reduced to two sentences. Thank you.

- in par. 3.2. SD values could be removed from the text (they are in the tables);

RESPONSE: We removed the SD values, as suggested. Thank you.

- some values are in the text but non presented in tables. In this case Authors could clarify why (e.g., lines 240-243; 263-267; 277-280);

RESPONSE: For lines 240-243, the years of experience are presented after level of education in the middle of Table 1. For lines 263-267, these were revised and specified that post hoc tests were further calculated (however, not included in the table). Similar situation happened for lines 277-280. For lines 263-267 and 277-280, we only included the narrative presentation in texts because presenting post hoc test results will add two tables. Kindly know that we are open to the suggestions of the honorable reviewer. Thank you.

- in Table 1 values regarding “Senior nursing specialist” are missing;

RESPONSE: We added its corresponding values. Thank you.

- par. 3.5.; I would suggest to replace “other” with “overall”, or “various” (or something similar), since both Professional Categories and Educational Levels have already been examined above;

RESPONSE: We replaced “other” with “various”. Thank you.

- Table 5: p values are not presented in the table. At least, it could be useful to put in bold (or to put an asterisk close to) the significant values in the Table. Otherwise it is difficult for the readers to follow the text explaining this table.

RESPONSE: P values have been added with asterisk for significant results. Thank you.

Discussion. This section is well done. I would only suggest some minor adjustments:

- lines 307-317: to consider a study carried out in Canada https://pubmed.ncbi.nlm.nih.gov/34806439/ (in 2015) reporting (for ASCOP) a higher overall item mean score of 4.8 (out of 6.0);

RESPONSE: This has been added, as suggested. Thank you.

- lines 313-317: for greater clarity, Authors could reformulate the sentence and first repeat the value of ASCOP in their study, i.e. 4.64 out of 6, which is greater than a previous study in Saudi Arabia, i.e. 4.59 out of 6. However, the fact that the latter was carried out in 2005 should be considered;

RESPONSE: We revised this in two sentences for better clarity. Thank you.

- lines 330-333: this sentence should be reformulated better.

RESPONSE: We revised this in two sentences for better clarity. Thank you.

Implications. Could be put before Conclusions.

RESPONSE: We transferred this, as suggested.

Conclusions. This section is consistent with the evidence presented.

RESPONSE: Thank you very much for the positive comment.

Informed Consent Statement. Authors could specify if this was signed online.

RESPONSE: We specified that it was signed online. Thank you.

Data Availability Statement. Authors could add this section to state, for instance, if data presented in this study are openly available in a repository, or available in another form. Or conversely, data are not available due to privacy issues, since information could compromise the privacy/anonymity of research participants, and so on.

RESPONSE: We added a statement about this. Thank you.

References are updated. However, these should be put in the right format (i.e., in the style requested by the Journal in the final reference list at the end of the paper).

RESPONSE: The numbering of references has been updated accordingly based on additional citations. Thank you very much for the positive comment.

English language is good. Minor editing of English language is required.

RESPONSE: Thank you very much for the positive comment.

Round 2

Reviewer 2 Report

Comments and Suggestions for Authors

Comments on the Quality of English Language

English language is good. Minor adjustments are required.

Author Response

Point-by-point Response to Reviewer 2 Comments – Second Round

Authors did a great and accurate work. Overall the suggested integrations and clarifications have been provided. I would only propose and motivate some final adjustments.

RESPONSE: Thank you so much for the positive feedback. We sincerely apologize if we did not completely fulfil some minor revisions. In the second round of revisions, we revised our paper based on your valuable comments for final adjustments.

Introduction.

  • I would suggest to integrate current par. 1.1., and sub-par. 1.1.1. and 1.1.2., at the end of the Introduction, as final sentence of this section (without providing the aim of the paper in a short paragraph).

RESPONSE: We revised these parts as suggested and also, in line with the original comments in the first round.  We hope that we understood your comments regarding this and also hope that we did the correct revisions. Due to these revisions, we also adjusted the numbering of the references (references 1–26), correspondingly. Thank you.

  • I suggested to add the number page of the lacking citation at line 67, current line 63. This since, regarding the block of lines 58-68, I saw that authors always cite the number page of the sentences reported, apart from this “the range of roles, functions, responsibilities and activities which a registered nurse is educated, competent and has authority to perform” [10]”. Thus I suggested to put also here the number page. In this respect, in their response, authors mention a reference that is the number 7 in their list. In the sentence I reported the ref. number 10 is cited. The number page of this citation is lacking, by my opinion. I’m sorry if I was not clear in my previous comment.

RESPONSE: Now, it is very clear, and we provided “p. 7”. Thank you.

  • Authors have put some full definitions as requested, but deleted the respective abbreviations, since they appear only in one part of the paper. By my opinion, despite this, the abbreviations could be put close to the full definitions, since in the literature the abbreviations are used and are useful for the readers. But this is only a suggestion. I’m referring to the following: (lines 80-83): Licensed Practical Nurses (LPNs), Registered Nurses (RNs) and Registered Psychiatric Nurses (RPNs).

RESPONSE: We agreed with the valuable opinion of the honorable reviewer and provided both full definitions and abbreviations. Thank you.

Results.

  • Regarding some data that are in the text but non presented in tables, I fully understand reasons provided by authors. However, they could specify this occurrence with a sentence close to them (e.g., data not presented in tables). This for greater clarity in the paper.

RESPONSE: We added this in lines 319-320 and 336, as suggested. Thank you.

Discussion.

  • Authors added the suggested reference (current n. 21), but it should be better formatted in the final list, i.e., Braithwaite; S.; Tranmer, J.; Wilson, R.; Almost, J.; Tregunno, D. Measuring Scope of Practice Enactment Among Primary Care Registered Nurses. Can. J. Nurs. Res. 2022, 54(4), 508-517. https://doi.org/10.1177/08445621211058328

RESPONSE: We corrected reference #21, as suggested for better formatting. Thank you so much.

  • By my opinion, the ASCOP score found in Saudi Arabia (4.59 out of 6) could also remain in the text. Also, maybe it could be useful to repeat the values of ASCOP in the study of the authors at line 312, e.g.: “The overall ASCOP mean score in our study (4.64 out of 6) is higher than a number of previous studies…..”

RESPONSE: We revised this, as suggested. Thank you.

  • I suggested to reformulate the sentence at lines 330-333. Authors however revised another sentence, not the one I suggested. I mentioned lines 330-333, that in the revised paper are new lines 342-345. A clearer sentence could be the following (it is only an example): “According to Dumit [31], the teaching of patient and family is a central role of nursing, and the majority of patient and family education in any health institution is provided by nurses. In this respect, nurses should train patient and family about the main aspects, mechanisms, benefits, and potential complications of each treatment procedure”

RESPONSE: We used this revision, as suggested for clearer sentence construction. Thank you.